# Cybersecurity Risk Assessment: A Systematic Mapping Review, Proposal, and Validation

**Isaac Daniel Sánchez-García** [1,*] **, Jezreel Mejía** [2] **and Tomás San Feliu Gilabert** [1]

1   Escuela Técnica Superior de Ingenieros Informáticos, Universidad Politécnica de Madrid (UPM), 28660 Madrid, Spain
2   Centro de Investigación en Matemáticas A.C., Zacatecas 98000, Mexico
*   Correspondence: isaacdaniel.sanchez@alumnos.upm.es

**Abstract:** Incorporating technologies across all sectors has meant that cybersecurity risk assessment is now a critical step in cybersecurity risk management. However, risk assessment can be a complicated process for organizations. Therefore, many authors have attempted to automate this step using qualitative and quantitative tools. The problems with the tools and the risk assessment stage in general are (1) not considering all the sub-steps of risk assessment and (2) not identifying the variables necessary for an accurate risk calculation. To address these issues, this article presents a systematic mapping review (SMR) of tools that automate the cybersecurity risk assessment stage based on studies published in the last decade. As a result, we identify and describe 35 tools from 40 primary studies. Most of the primary studies were published between 2012 and 2020, indicating an upward trend of cyber risk assessment tool publication in recent years. The main objectives of this paper are to: (I) identify the differences (reference models and applications) and coverage of the main qualitative and quantitative models, (II) identify relevant risk assessment variables, (III) propose a risk assessment model (qualitative and quantitative) that considers the main variables and sub-stages of risk assessment stage, and (IV) obtain an assessment of the proposed model by experts in the field of cybersecurity. The proposal was sent to a group of 28 cybersecurity experts who approved the proposed variables and their relevance in the cybersecurity risk assessment stage, identifying a majority use of qualitative tools but a preference of experts for quantitative tools.

**Keywords:** risk assessment; automation; tools; algorithms; cybersecurity; systematic mapping review; proposal; experimentation; validation; cybersecurity experts

## 1. Introduction

The current interaction among organizations, people, and cyberspace has resulted in the need to identify and manage the risks associated with using cyberspace. Cybersecurity transcends the limits of traditional information security to include not only the protection of information resources but also the protection of other assets, including people themselves [1].

Companies needing to identify and manage risks related to cybersecurity have adopted risk management models. All models for managing cybersecurity risks generally involve stages. The number and name of the stages vary depending on the model, although the risk assessment stage is the most used by organizations and is present in all risk management models [2].

Risk assessment is the part of the risk management process that incorporates the analysis of threats and vulnerabilities. Additionally, risk assessment considers the mitigations provided by planned or implemented security controls [3]. Thus, risk assessment is a crucial stage for performing a correct cybersecurity risk management process. Some examples of risk assessment models are the one proposed by the International Organization for Standardization (ISO) in ISO/IEC 27005:2018 [4], the one proposed by the National

Institute of Standards and Technology (NIST) in the NIST Cyber Security Framework [5], and the one created by Carnegie Mellon University (USA), OCTAVE [6].

Forbes pointed out that one of the future challenges of risk assessment will be performing correct automation [7]. Risk assessment is usually automated based on the risk assessment models developed by international organizations such as those mentioned above.

There are several problems derived from the automation and implementation of a risk assessment model, such as, for example:

- Selection and use of inappropriate tools that do not mitigate the predominant type of risk in their industry [8];
- Implementation of a partly or incomplete automated risk assessment [9];
- Use of tools that focus on information security risks that may not adequately assess cybersecurity risks [1];
- Consideration of using a quantitative risk calculation tool where the maturity level is inadequate and vice versa [9].

The present research work aims to solve the previous problems by means of achieving the following objectives: (I) identify the differences (reference models and applications) and coverage of the main qualitative and quantitative models, (II) identify relevant risk assessment variables, (III) propose a risk assessment model (qualitative and quantitative) that considers the main variables and sub-stages of risk assessment stage, and (IV) obtain an assessment of the proposed model by experts in the field of cybersecurity.

The first problem relating to the achievement of the objectives proposed above is the lack of literature on risk assessment in cybersecurity. Therefore, for objectives I, II, and III, we proposed to obtain and characterize information on the quantitative and qualitative risk assessment models used by the tools by means of a systematic mapping review (SMR). To meet objective IV, a questionnaire was designed to collect demographics, risk management practices, opinions on risk assessment practices, and an evaluation of the risk assessment proposal. This survey was sent to cybersecurity experts to obtain an objective evaluation.

The key contributions of this research work can be summarized as: (1) identifying cybersecurity risk assessment tools, (2) knowing where risk assessment tools and models are applied, (3) validating which part of the risk assessment stage is covered within these tools, (4) analyzing the variables involved in the sub-stages and activities of the risk assessment stage, and (5) proposing and validating a quantitative and qualitative model to solve the deficiencies identified in the SMR.

This article is divided into seven sections. Section 1 is the introduction, highlighting the importance of the risk assessment stage, its automation, and its problems. Section 2 addresses the background and describes the risk assessment stage. Section 3 presents the SMR (systematic mapping review) process and its procedures. Section 4 reports and analyzes the SMR results. Section 5 presents a proposal to assess cybersecurity risk. Section 6 describes the validation survey and the results to evaluate the proposal presented in Section 5. Finally, Section 7 outlines the conclusions and future work.

## 2. Background

Before characterizing and comparing risk assessment tools, we must first characterize the risk assessment stage. This section provides an overview of the concepts and steps of the risk assessment stage based on the ISO/IEC 27000 family and considers other models such as NIST 800-30, NISTCSF, IRAM, OCTAVE, and OWASP.

The structure of the ISO/IEC 27000 family risk assessment stage was taken as a reference model for the comparison because:

- The ISO/IEC 27000 standard family is considered by Stoll [10] as the most important standard for risk assessment;
- According to Susanto et al. [11], the ISO 27000 family is one of the most important and accepted international initiatives for developing and operating a cybersecurity and information security management system (ISMS);

- The 2020 Assessment System Standard Certifications Survey [12] states that the ISO/IEC 27000 family is one of the International Organization for Standardization standards most certified by companies.

Additionally, the ISO/IEC 27032:2012 guidelines for cybersecurity [13] mention that ISO 270032 cybersecurity risk assessment sub-stages are the same as those of the ISO/IEC 27005:2018 standard.

The sub-stages may vary from model to model. However, most models share essential risk assessment activities, such as (1) identification of assets, (2) identification and evaluation of threats, (3) identification and evaluation of vulnerabilities, and (4) risk measurement. The ISO/IEC 27005 risk assessment model [4] was selected to standardize the concepts and content of each sub-stage.

The risk assessment stage of the ISO/IEC 27005:2018 standard (see Figure 1) is divided into three main sub-stages. These sub-stages are:

- Risk identification [4]: This sub-stage aims to determine the possible causes of a potential loss and gain insight into how, where, and why the loss can happen. This sub-stage is divided into the following activities:
  - Identification of assets: related to information, process, or people the organization is interested in protecting;
  - Identification of existing controls: related to implemented controls identification;
  - Identification of vulnerabilities: related to weaknesses, failures, or deficiencies of an asset that may generate a risk for the asset;
  - Identification of threats: related to everything which may exploit a vulnerability or have an interest in the asset;
  - Identification of consequences: related to confidentiality, integrity, and availability;
- Risk analysis [4]: This sub-stage identifies the risk analysis methodologies, techniques, models, or guides required to carry out an adequate risk estimation. Risks can be estimated using either quantitative or qualitative methods. The risk analysis sub-stage is divided into the following activities:
  - Assessment of the consequences: to assess the potential business impact of possible or actual information security incidents for the organization, considering the consequences of a breach of information security, such as loss of confidentiality, integrity, or availability of the assets;
  - Assessment of incident likelihood: related to assessing the likelihood of the incident scenarios;
  - Level of risk determination: related to determining the risk level for all relevant incident scenarios;
- Risk evaluation [4]: This sub-stage compares an acceptable or tolerable risk value defined in the context establishment stage (risk evaluation and acceptance criteria) with the risk analysis sub-stage values. In this way, the organization can assess the priority of each risk.

Among the sub-steps that make up the risk assessment stage, the one with the most significant activities is risk identification. The risk identification sub-stage includes fundamental risk assessment activities, such as identifying vulnerabilities and threats [14]. Likewise, authors such as Northern et al. [15] highlight the relevance of both the vulnerability assessment (risk identification sub-stage) and the risk determination (risk analysis). Level of risk determination, being a probabilistic term based on impact, helps to determine the relevance or priority of a risk. Finally, the risk evaluation sub-stage is essential to completing the continuous improvement of a risk management system. Residual risk should be compared to the risk acceptable to the organization to identify the maturity and current state of an organization's security risk [16].

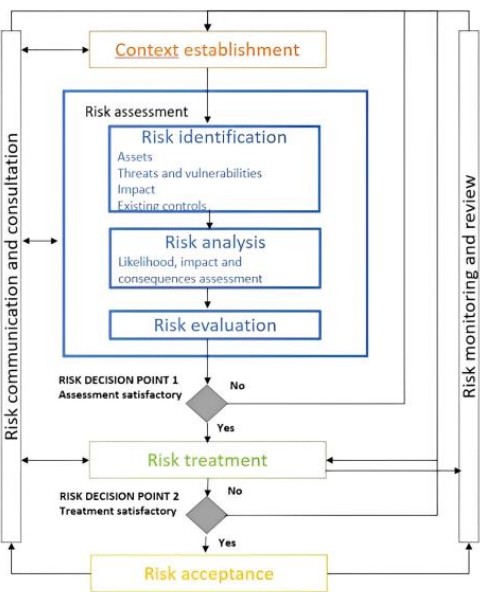

**Figure 1.** ISO/IEC 27005:2018 risk management model (extracted from the standard [4]).

Not contemplating any of the sub-steps mentioned above or activities may cause a loss of objectivity. Furthermore, the objectives of an adequate risk assessment would not be achieved either. It is also important that, when automating the risk assessment, all of these sub-steps and activities are considered.

After defining the risk assessment sub-stages and activities, we could compare the tools that automate risk assessment. The following section describes the SMR process wherein the main risk assessment tools were identified.

## 3. Systematic Mapping Review (SMR) Process

The proposed SMR aimed to identify the primary studies related to tools that automate the cybersecurity risk assessment stage. Petersen et al.'s methodology [17] was selected to conduct the SMR. A SMR is a process that applies a repeatable process to search and analyze studies and provides descriptive information about the state of the art of a topic and a summary of the conducted research [18]. This SMR included the following steps: (1) proposal of a review protocol and definition of research questions, (2) performance of the review (identifying and evaluating primary studies), (3) extraction of the relevant information, and (4) discussion and analysis of the results. This section details Steps (1), (2), and (3). Section 4 describes SMR Step (4).

One of the main challenges of this SMR was the large number of articles analyzed as a result of the search in the "Springer Link" and "IEEE digital library" search engines. In order to solve this problem, this SMR was carried out with the help of the web tool "Parsifal" (https://parsif.al/, accessed on 25 December 2021).

Parsifal is one of the most used tools by researchers to carry out systematic reviews and mapping in software engineering [19]. One of the most significant advantages of Parsifal is that the tool supports the collaboration of its users and allows multiple users to work on an SMR [20]. Parsifal also includes a semi-automatic search, where defined search strings automatically access databases (Bustos Navarrete et al., 2018). This tool manages search engine results more efficiently than a traditional spreadsheet.

### 3.1. Propose a Review Protocol and Define Research Questions

To achieve the objectives I, II, and III of this research work, it was necessary to define a set of questions called research questions in this SMR. We formulated four research questions to classify and retrieve relevant information from the studies related to tools that automate the cybersecurity risk assessment stage.

Research Question 1 (RQ1): What tools are mentioned in the literature on cybersecurity risk assessment automation? The response to RQ1 will identify the tools that automate the cybersecurity risk assessment reported in the literature. They will also be classified according to their focus (cybersecurity or information security) to identify the universe of tools applicable to cybersecurity. This research question will help to achieve objective I by identifying which risk assessment models automations take as a reference.

Research Question 2 (RQ2): What cybersecurity risk assessment automation applications are mentioned in the literature? The response to RQ2 will identify the risk type and focus area addressed by the tools. This research question will help to achieve objective I, identifying applications of the automations of the risk assessment stage.

Research Question 3 (RQ3): What sub-stages of the cyber risk assessment stage are mentioned in the literature? Some tools will mention whether they cover the entire risk assessment stage or only certain specific sub-stages. The response to RQ3 could determine which tools automate a complete or partial cybersecurity risk assessment stage. This research question will help to achieve objective I, identifying the coverage of each of the automations.

Research Question 4 (RQ4): What is the main basis of risk measurement used by the tools (mentioned in the literature) to assess cybersecurity risks? The response to RQ4 will identify the type of risk measurement mentioned by the literature and performed by tools (quantitative or qualitative). This research question will help to achieve objectives II and III, identifying and analyzing the different variables used in qualitative and quantitative models.

### 3.1.1. Criteria for the Selection of Sources

The source selection criteria were: (1) databases that include journals and studies focused on cybersecurity risk assessment automation and have advanced search mechanisms that make use of the terms and synonyms used in search queries and provide access to the full text of studies; (2) studies available on the web for free; and (3) specialized magazines available via the Universidad Politecnica de Madrid's library.

### 3.1.2. PICO Protocol

This SMR was conducted by consulting the ACM Digital Library, IEEE Digital Library, ISI Web of Science, Science Direct, and Springer Link digital libraries using the PICO protocol (population, intervention, comparison, and outcome).

- Population: Publications related to automation (tools) of cybersecurity or the information security risk assessment stage (used for cybersecurity purposes);
- Intervention: Type of automation, in this case, cybersecurity or information security risk assessment tools (used for cybersecurity purposes);
- Comparison: Comparative studies of tools are considered. Additionally, the different tools retrieved are compared for characterization purposes;
- Outcome: Studies of cybersecurity or information security risk assessment tools (used for cybersecurity purposes).

### 3.1.3. Research String Generation

A set of keywords was selected to retrieve the highest number of results. In addition, synonyms for the keywords were included to avoid excluding relevant studies and to efficiently build the search string. The keywords used and their synonyms are shown in Table 1.

The search strings were built using the keywords and synonyms from Table 1. Then, the logical connectors "AND" and "OR" were added to join the keywords and synonyms, constructing the following generic search string: ("Tool" OR "Algorithm") AND ("cybersecurity" OR "Cyber security" OR "Information security") AND ("Risk assessment" OR "Risk Management" OR "Risk evaluation").

The sources were searched using the criteria defined in their selection (see Section 3.1.1). All identified database sources were included. Search strings were applied to electronic databases and other sources (journals and conferences).

**Table 1.** Keywords and synonyms.

| Keywords | Synonyms |
|---|---|
| Cybersecurity | Cyber security |
| Information security | |
| Risk assessment | Risk management, Risk evaluation |
| Tool | Algorithm |

### 3.2. Conduct the Review

The search string defined in the previous step was entered into the search engines, and inclusion and exclusion criteria were created.

### 3.2.1. Inclusion and Exclusion (I and E) Criteria

The primary study selection process was conducted using the following inclusion and exclusion criteria:

(a)  Inclusion criteria:

- Free-access studies;
- Studies comparing risk assessment tools;
- Studies where the title, abstract, and keywords are related to the topic;
- Complete studies;

(b)  Exclusion criteria:

- Duplicate studies;
- Studies based only on a particular opinion that does not address cybersecurity risk assessment;
- Studies that do not mention what criteria are used to assess cybersecurity risks;
- Studies published prior to 2010, due to the rapid evolution of the cybersecurity field;
- Studies that are irrelevant to the research questions or not related to the topic;
- Unclear or ambiguous studies;
- Studies that mention tools that are not used in cybersecurity risk assessment;
- Studies without any automation of the risk assessment stage;
- Gray literature, or literature published by non-traditional publishers.

### 3.2.2. Selection of Primary Studies

The selection of primary studies was a procedure consisting of four activities. The details of each activity and the analyzed documents can be found in the following document: https://short.upm.es/ytaub (accessed on 15 February 2022). In the first activity, we inserted the search string created in the review protocol into the database search engines. After the first screening, 544 studies were found. The results are shown in Table 2.

**Table 2.** Digital libraries and search string results.

| Sources | Results |
|---|---|
| ACM Digital Library | 14 |
| IEEE Digital Library | 111 |
| ISI Web of Science | 96 |
| Science@Direct | 37 |
| Springer Link | 286 |
| Total | 544 |

We used the Parsifal tool to perform the second activity, where the search engine results were imported in BibTex format (file or text). Each of the studies was reviewed considering the inclusion and exclusion criteria.

After applying the inclusion and exclusion criteria to the title, abstract, and keywords of all 544 studies, 397 were excluded because they were irrelevant to the research questions. Additionally, we found 44 duplicate studies, eight pre-2010 studies, and 30 studies based on a single opinion. Six studies did not mention the criteria used for risk assessment, six did not mention any tool related to the risk assessment stage, and two needed to be clarified. A total of 51 publications were accepted. The complete list of accepted studies is shown in Appendix A "Relevant studies before quality assessment" or in the following document: https://short.upm.es/ytaub (accessed on 18 August 2022).

We read the full text in the third activity to select relevant studies. Studies that provided enough information were selected and saved. After applying the third activity, 11 studies were rejected, and we accepted 40.

Finally, in the fourth activity, we applied a quality questionnaire consisting of five quality assessment (QA) questions created to validate the quality of the 40 accepted studies. The five questions are listed as QA1 through QA5 in Table 3. Table 3 also shows the questions and how they were evaluated.

**Table 3.** Questions and criteria for quality assessment.

| | Question | Yes 1.0 | Partially 0.5 | No or Not Mentioned 0.0 |
|---|---|---|---|---|
| QA1 | Does the study mention the model on which the tool is based? | The underlying model is mentioned directly | The underlying model is mentioned without details | The underlying model is not mentioned |
| QA2 | Is the tool free of charge? | The mentioned tool is free of charge | N/A | The tool is licensed, for subscribers, or not mentioned |
| QA3 | Does the study have information about tools for information security or cybersecurity risk assessment? | The study mentions that the tool focuses on cybersecurity | The study mentions that the tool is focused on information security | The study is not focused on information security or cybersecurity or does not specify its focus |
| QA4 | Does the study mention how the tool performs the risk assessment stage? | It is clearly and precisely stated how the tool performs the risk assessment stage | Risk assessment performance is not mentioned in detail or is incomplete | It is not mentioned |
| QA5 | Does the study mention the sub-stages of risk assessment that it covers? | The study states the sub-stages of the risk assessment stage covered by the tool | The study only describes certain sub-stages of the risk assessment stage covered by the tool | No risk assessment sub-stages are automated in the study or they are not mentioned |

First, there were three potential responses to each question used to evaluate the studies: "YES", which assigned a value of 1.0 to the study; "Partially", which assigned a value of 0.5 to the study; and "NO", which assigned a value of 0.0 to the study. Second, each quality question was applied to the 40 accepted studies mentioned above. Each study's value (score) was in the range of 0.0 to 5.0. Finally, it was established for the quality assessment that only studies with scores equal to or greater than 2.5 would be considered primary studies.

Of the 40 studies, only 26 had a score greater than 2.5. The studies are listed in Table 4. We extracted the relevant information from the 26 studies selected after applying the quality assessment to answer the research questions.

**Table 4.** Primary studies.

| ID | Title | Year | Authors |
|----|-------|------|---------|
| [21] | Asset Assessment in Web Applications | 2010 | Romero, M. and Haddad, H. |
| [22] | A visualization and modelling tool for security metrics and measurements management | 2011 | Savola, R. and Heinonen, P. |
| [23] | Introduction of a Cyber Security Risk Analysis and Assessment System for Digital I&C Systems in Nuclear Power Plants | 2013 | Lee, C. |
| [24] | Sector-Specific Tool for Information Security Risk Management in the Context of Telecommunications Regulation (Tool Demo) | 2014 | Mayer, N. and Aubert, J. |
| [25] | Experimentation tool for critical infrastructures risk management | 2015 | Bialas, A. |
| [26] | Smart grid cybersecurity risk assessment | 2015 | Langer, L. et al. |
| [27] | Security Assessment of Information System in Hospital Environment | 2016 | Tritilanunt, S. et al. |
| [28] | A risk assessment model for selecting cloud service providers | 2016 | Cayirci, E. et al. |
| [29] | A Comparison of Cybersecurity Risk Analysis Tools | 2017 | Roldán-Molina, G. et al. |
| [30] | Business Driven ICT Risk Management in the Banking Domain with RACOMAT | 2017 | Viehmann, J. |
| [31] | Open-source intelligence for risk assessment | 2018 | Hayes, D. and Cappa, F. |
| [32] | Mobile Information Security Risk Calculator | 2019 | Tukur, Y. |
| [33] | Reducing Informational Disadvantages to Improve Cyber Risk Management | 2018 | Shetty, S. et al. |
| [34] | Audit Plan for Patch Management of Enterprise Applications | 2018 | Odilinye, L. et al. |
| [35] | Introduction of a Tool-based Continuous Information Security Management System: An Exploratory Case Study | 2018 | Brunner, M. et al. |
| [36] | RL-BAGS: A Tool for Smart Grid Risk Assessment | 2018 | Wadhawan, Y. et al. |
| [37] | Security risk situation quantification method based on threat prediction for multimedia communication network | 2018 | Hu, H. et al. |
| [38] | CSAT: A User-interactive Cyber Security Architecture Tool based on NIST-compliance Security Controls for Risk Management | 2019 | Huang, Y. et al. |
| [39] | A Web Platform for Integrated Vulnerability Assessment and Cyber Risk Management | 2019 | Russo P et al. |
| [40] | Development of Threat Modelling and Risk Management Tool in Automated Process Control System for Gas Producing Enterprise | 2019 | Rimsha, A. and Rimsha, K. |
| [41] | Automatic network restructuring and risk mitigation through business process asset dependency analysis | 2020 | Stergiopoulos, G. et al. |
| [42] | Leveraging cyber threat intelligence for a dynamic risk framework | 2019 | Riesco, R. and Villagrá, V. |
| [43] | I-HMM-Based Multidimensional Network Security Risk Assessment | 2020 | Hu, J. et al. |
| [44] | Calculated risk? A cybersecurity evaluation tool for SMEs | 2020 | Benz, M. and Chatterjee, D. |
| [45] | Tackle Cybersecurity and AWIA Compliance with AWWA's New Cybersecurity Risk Management Tool | 2020 | Ohrt, A. et al. |
| [46] | Algorithm for quickly improving quantitative analysis of risk assessment of large-scale enterprise information systems | 2020 | Teng, Y. et al. |

### 3.3. Extract the Results

A template was designed to extract the important information from each study. The template contained the fields shown in Table 5. Next, we read the full text for each selected study and recorded the information in the template. This allowed for the subsequent analysis of the results.

**Table 5.** Data extraction form.

| Data Item(s) | Descriptions |
|--------------|--------------|
| ID reference | Study identifier |
| Reference | Title, author(s), year, and publication venue |
| Tools | Mentioned and explained by the author |
| Reference models | Risk assessment models mentioned by the author |
| Purpose | The cybersecurity focus area of the study that the author mentions |
| Sub-stages | Risk assessment sub-stages that the tools automate |
| Type of risk measurement | How the tool assesses risk |

## 4. Analyze and Discuss the Results

This section describes the SMR results based on the data extracted from the 26 primary studies. The year of publication of the studies helped us to identify trends related to efforts to automate the cybersecurity risk assessment stage. For example, Figure 2 shows that the number of studies mentioning tools that automate the cybersecurity risk assessment

stage increased as of 2018. The trend to create more automation responds to the need for companies to invest less effort and fewer resources in their implementation [7].

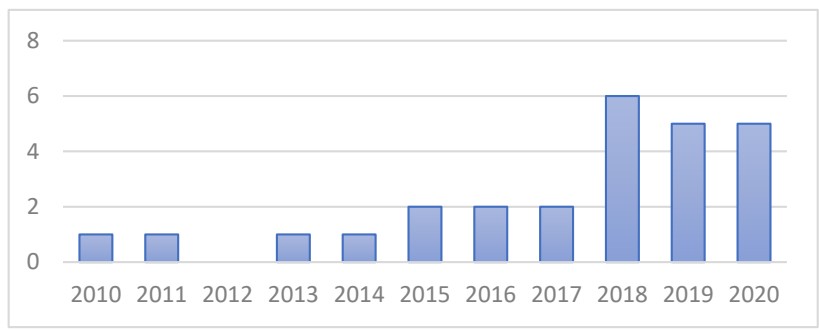

**Figure 2.** The number of studies per year.

The positive trend indicates that cybersecurity risk assessment is a topic that constantly continued after 2018, with more publications than before 2017.

Figure 3 analyzes the type of publication in the literature. We found that publications were mostly concentrated in specialized journals and conference proceedings. This trend is because the tools, being direct applications of models, tend to have an impact of greater interest to the organizations than the models themselves. This trend is consistent with the potential relevance of tools that automate cybersecurity risk assessment.

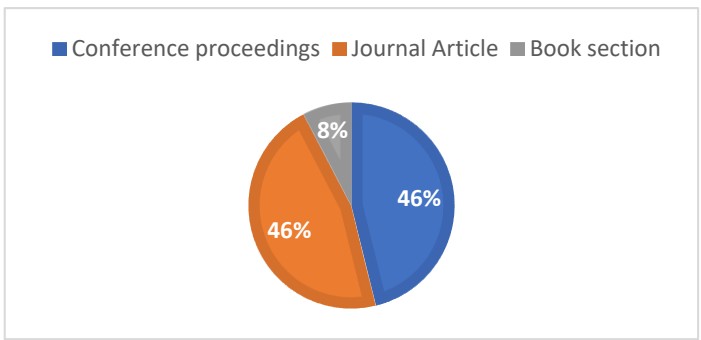

**Figure 3.** Percentage of studies per their type.

The information extracted from the primary studies that are analyzed and discussed in this section is related to the research objectives and RQs as follows: (1) cybersecurity risk assessment tools (reference models and applications) are related to RQ1 and RQ2 and research objective I, (2) coverage of tools is related to RQ3 and research objective I, and, finally, (3) risk measurement is related to RQ4 and research objectives II and III.

*4.1. Cybersecurity Risk Assessment Tools (Reference Models and Applications)*

Of the 26 primary studies analyzed, we identified 35 tools which focus on automating cybersecurity risk assessment. We grouped the tools that take a model as a reference, identifying 20 tools. As can be seen in Table 6, both qualitative and quantitative tools are based on risk assessment models. The relevance of identifying the risk assessment models on which the tools are based helped us to perform the characterization in Section 4.3 of which variables and types of calculations are performed by risk assessment tools focused on cybersecurity.

These tools take a reference model to facilitate integration with international standards. The reference models most used by the tools are the ISO/IEC 27000 family, the NIST 800-30 model, and the NIST Cyber Security Framework, which rounds out the NIST 800-30 model with cybersecurity risks.

**Table 6.** Risk assessment tools with an underlying risk assessment model extracted from primary studies.

| Tool Name | Base Model | Type of Tool |
|---|---|---|
| ADAMANT [35] | ISO/IEC 27001, NIST 800-30 | Qualitative |
| OSCAD-Ciras tool [25] | ISO/IEC 27001, ISO/IEC 31000 | Qualitative |
| SGIS toolbox [26] | ISO/IEC 27001, NIST 800-30 | Qualitative |
| MST risk analysis [41] | ISO/IEC 27005, NIST 800-30, Octave | Quantitative |
| TISRIM [24] | ISOIEC 27005 | Qualitative |
| Teng algorithm [46] | ISO/IEC 27005 | Quantitative |
| R. Riesco model [42] | ISO/IEC 27000 series | Quantitative |
| Mobile risk calculator [32] | NIST 800-30, RISK watch, CORA | Quantitative |
| Nessus [27,29,43] | NIST 800-30, HIPAA, PCI | Qualitative |
| CSRAS [23] | NIST 800-30 | Qualitative |
| CYRVM [39] | NIST 800-30 | Quantitative |
| Cybersecurity evaluation tool CSET [44] | NIST CSF, ISO/IEC 27001 | Qualitative |
| Patch management [47] | NIST CSF, ISO/IEC 27001 | Qualitative |
| AWWA [45] | NIST CSF | Qualitative |
| CSAT [38] | NIST CSF | Qualitative |
| Asset assessment methodological tool [21] | EBIOS, MAGERIT | Qualitative |
| RACOMAT [30] | ISO/IEC 31000 | Qualitative |
| Data protection impact assessment tool [28] | CARAM | Quantitative |
| MVS visualization platform [22] | ITSEC | Qualitative |
| Tritilanunt, Suratose tool [27] | OWASP | Qualitative |

The tools designed to comprehensively cover the risk assessment stage take an existing and accepted risk assessment model as a reference. The reference model also influences whether the tool is qualitative or quantitative. As we can see in Table 6, most of the tools that take as reference the ISO 27000 family or NIST 800-30 have a quantitative evaluation method.

From Table 6, we can conclude that a relevant factor in security risk assessment tools is that the tools seek coupling with international certifiable models such as the ISO 27000 family. In addition, the base calculation type is usually of the same type as the reference model.

Another group of tools was created with a different goal to carrying out a risk assessment (which is why no reference model is mentioned). For example, Eye retina, GFILanguard, and N circle aim to identify vulnerabilities in a specific system or technology, such as system configurations or logs. Despite having been created to identify technical vulnerabilities, these tools can be used to cover certain sub-stages of the risk assessment stage. Another example is OSINT (open-source intelligence tools), which focus on extracting information from public sources, such as web pages or social networks. OSINT information is used to perform cybersecurity intelligence analysis. Their primary purpose is not directly focused on the risk assessment stage, but they can be used to perform social engineering analysis to identify risks.

Another important characteristic is the type of risk targeted by the tools. Some examples are tools which focus on carrying out a cybersecurity risk assessment to cover the network or cloud domain. As a result, targeted tools may need to correctly address risks from other cybersecurity domains.

For each tool, we identified the type of risk on which it is focused (Type of Risks column in Table 7) and the type of sector or industry it targets (Focus Area column in Table 7). In addition, the name of the tool is added in column 1.

In addition, 25 out of 35 studies addressed general-purpose tools applicable to all kinds of organizations. General-purpose tools are the most interesting since they can be applied to any industry or sector. However, some identified tools are specific to a sector. For example, the Tritilanunt tool focuses on the health sector; the OSCAD-Ciras tool focuses on a nation's critical infrastructure (water, gas, and communications services). The TISRIM

tool is focused on the telecommunications sector, and the SGIS toolbox is focused on the energy sector.

**Table 7.** Tools related to the type of risk and focus area.

| Tool | Type of Risks | Focus Area |
|---|---|---|
| ADAMANT | Any | General purpose |
| AWWA | Any | Water infrastructure |
| CSAT | Any | Information systems |
| CSET | Any | General purpose |
| CSRAS | Any | Banks |
| MVS visualization platform | Any | General purpose |
| OSCAD-Ciras tool | Any | Critical infrastructure |
| OSINT | Any | Critical infrastructure |
| Patch management | Any | General purpose |
| RACOMAT | Any | General purpose |
| RL-BAGS | Any | Critical infrastructure |
| Risk watch | Any | General purpose |
| Teng algorithm | Any | General purpose |
| TISRIM | Any | Telecommunications |
| Tritilanunt, Suratose tool | Any | Health |
| Mobile risk calculator | Any, except human risks | General purpose |
| Eye retina | Technical risk | General purpose |
| GFILanguard | Technical risk | General purpose |
| N circle | Technical risk | General purpose |
| Nessus | Technical risk | General purpose |
| Nikto | Technical risk | General purpose |
| OpenVAS | Technical risk | General purpose |
| Qualys guard | Technical risk | General purpose |
| SAINT | Technical risk | General purpose |
| Security system analyzer | Technical risk | General purpose |
| CYRVM | Network risk | General purpose |
| MST risk analysis | Network risk | General purpose |
| J. Hu model | Network risk | General purpose |
| SGIS toolbox | Network risk | Energy distribution |
| Asset assessment methodological tool | Web application development risk | General purpose |
| CRISM tool | Software vulnerabilities and network | General purpose |
| Data protection impact assessment tool | Risks in the cloud | General purpose |
| Quantitative network security risk situation model | Multimedia communication traffic risk | General purpose |
| R. Riesco model | Cyber-attack risks | General purpose |
| Rimsha tool | Software risk | Gas industry |

A total of 15 of 35 tools can be considered to target all types of risks. Nineteen of the tools are focused on a specific type of risk. For example, the data protection impact assessment tool only focuses on risk assessment for cloud services relating to account hijacking or insider threats. Another example is the R. Riesco model, which is focused on risks derived from the cyber-defense field, such as phishing or watering hole attacks, and can be considered for any sector or industry since it is general purpose. A third example is Tritilanunt, Suratose's tool that, despite being focused on all types of risks, and having an application in the medical sector, emphasizes risks related to information privacy and exposure of personal information. Likewise, many vulnerability analysis tools such as Eye retina, Nessus, Nikto, or N circle are focused on more technical risks such as denial-of-service, man-in-the-middle, or password attacks that can affect all types of organizations in the same manner. Finally, there are tools, such as CYRVM, MST risk analysis, J. Hu model, or SGIS toolbox, that focus on risks related to networks and that seek to identify risks related to packet sniffing attacks, ping sweeps, and port scanning.

Summarizing the results of this subsection: (1) We can see that the ISO 27000 family and the NIST 800-30 are risk assessment models used as reference for qualitative and

quantitative calculations. (2) Models such as NIST CSF or OWASP are usually used for qualitative calculations. (3) An important group of tools does not use a reference risk assessment model. (4) Most tools are focused on any type of risk and/or are general purpose. (5) Another regressive group comprises tools that consider technical risks such as denial-of-service, man-in-the-middle, or password attacks.

*4.2. Coverage of Tools*

The risk assessment was divided into three sub-stages and six activities to compare the tools identified in the primary studies. These sub-stages and activities were selected based on the ISO/IEC 27005:2018 standard.

- Risk identification sub-stage:
  ○ Activity 1 was defined as identifying of the assets that must be part of critical business or organizational processes;
  ○ Activity 2 was defined as the identification of the threats and the identification of existing controls;
  ○ Activity 3 was defined as the identification of vulnerabilities and the identification of consequences;
- Risk analysis sub-stage:
  ○ Activity 4 was defined as the assessment of the consequences and the assessment of the incident likelihood;
  ○ Activity 5 was defined as the level of risk determination. It is the most important activity of the entire risk assessment stage;
- Risk evaluation sub-stage:
  ○ Activity 6 was defined as the selection of an acceptable risk value. The acceptable or tolerable risk value defined in the context establishment stage (see ISO/IEC 27005:2018) was compared with the risk value obtained in Activity 5, and each risk was prioritized.

The activities covered by the tools are shown in Table 8. Tools are listed in descending order by coverage. As a result, we identified 11 tools that cover all the risk assessment activities (and, thus, sub-stages).

The most common activities covered by the risk assessment tools were Activity 5 (define a risk value), which was covered by all tools, and Activity 3 (vulnerability identification), which was covered by all the tools, except for the asset assessment methodological tool. Regardless of their focus, all risk assessment models identify the vulnerabilities to perform a cybersecurity risk assessment. Therefore, these two activities are the most common and relevant in cybersecurity risk assessment tools.

When implementing a cyber risk assessment tool, it is necessary to identify which sub-stage or activity is targeted. Identifying sub-stage and activity coverage shows how to address the risks. Using a tool that only partially covers the risks can create a false sense of security certainty.

Summary of the results of this subsection: (1) Eleven tools covered all the sub-steps of risk assessment. (2) Activity 5 (risk determination) was the most covered by the tools. (3) Activity 3 (vulnerability assessment) was the second most relevant to the tools. (4) Activity 6 (comparison of acceptable risk value and calculated risk) was the least considered by risk assessment tools.

**Table 8.** Sub-stages and activities covered by the risk assessment tools.

| Sub-Stages | Risk Identification | | | Risk Analysis | | Risk Evaluation | |
|---|---|---|---|---|---|---|---|
| Tool | Activity 1 | Activity 2 | Activity 3 | Activity 4 | Activity 5 | Activity 6 | Coverage |
| ADAMANT | ✓ | ✓ | ✓ | ✓ | ✓ | ✓ | Full |
| CYRVM | ✓ | ✓ | ✓ | ✓ | ✓ | ✓ | Full |
| Data protection impact assessment tool | ✓ | ✓ | ✓ | ✓ | ✓ | ✓ | Full |
| Mobile risk calculator | ✓ | ✓ | ✓ | ✓ | ✓ | ✓ | Full |
| OSCAD-Ciras tool | ✓ | ✓ | ✓ | ✓ | ✓ | ✓ | Full |
| RACOMAT | ✓ | ✓ | ✓ | ✓ | ✓ | ✓ | Full |
| Risk watch | ✓ | ✓ | ✓ | ✓ | ✓ | ✓ | Full |
| R. Riesco model | ✓ | ✓ | ✓ | ✓ | ✓ | ✓ | Full |
| SGIS toolbox | ✓ | ✓ | ✓ | ✓ | ✓ | ✓ | Full |
| TISRIM | ✓ | ✓ | ✓ | ✓ | ✓ | ✓ | Full |
| Tritilanunt, Suratose tool | ✓ | ✓ | ✓ | ✓ | ✓ | ✓ | Full |
| J. Hu model | ✓ | Partial | Partial | ✓ | ✓ | ✓ | Partial |
| MVS Visualization Platform | ✓ | Partial | ✓ | ✓ | ✓ | ✓ | Partial |
| RL-BAGS | ✓ | ✗ | ✓ | ✓ | ✓ | ✓ | Partial |
| Patch management | Partial | ✓ | ✓ | ✓ | ✓ | ✗ | Partial |
| Cyber risk scoring and mitigation (CRISM) tool | Partial | Partial | ✓ | ✓ | ✓ | ✗ | Partial |
| Rimsha tool | Partial | Partial | ✓ | ✓ | ✓ | ✗ | Partial |
| MST | Partial | Partial | ✓ | ✓ | ✓ | ✗ | Partial |
| AWWA | ✗ | ✓ | ✓ | ✗ | ✓ | ✓ | Partial |
| Asset assessment methodological tool | ✓ | ✗ | ✗ | ✓ | ✓ | ✓ | Partial |
| Eye retina | Partial | ✗ | ✓ | ✓ | ✓ | ✗ | Partial |
| GFILanguard | Partial | ✗ | ✓ | ✓ | ✓ | ✗ | Partial |
| Nikto | Partial | ✗ | ✓ | ✓ | ✓ | ✗ | Partial |
| OSINT | Partial | ✗ | ✓ | ✓ | ✓ | ✗ | Partial |
| OpenVAS | Partial | ✗ | ✓ | ✓ | ✓ | ✗ | Partial |
| Quantitative network security risk situation | ✗ | Partial | ✓ | ✓ | ✓ | ✗ | Partial |
| SAINT | Partial | ✗ | ✓ | ✓ | ✓ | ✗ | Partial |
| N circle | Partial | ✗ | ✓ | ✓ | ✓ | ✗ | Partial |
| Qualys guard | Partial | ✗ | ✓ | ✓ | ✓ | ✗ | Partial |
| Security system analyzer | Partial | ✗ | ✓ | ✓ | ✓ | ✗ | Partial |
| Teng algorithm | ✗ | Partial | Partial | ✓ | ✓ | ✗ | Partial |
| CSAT | ✗ | ✓ | ✓ | ✗ | ✓ | ✗ | Partial |
| CSET | ✗ | Partial | ✓ | ✗ | ✓ | ✓ | Partial |
| CSRAS | ✓ | ✗ | ✓ | ✗ | ✓ | ✗ | Partial |
| Nessus | Partial | ✗ | ✓ | ✗ | ✓ | ✗ | Partial |

### 4.3. Risk Measurement

The 35 tools identified above perform risk measurement in two ways: qualitatively or quantitatively. Of the 35 tools, 23 perform a qualitative evaluation, and 12 perform a quantitative evaluation.

#### 4.3.1. Qualitative Tools

The qualitative tools (23 in total) are shown in Table 9. They provide ordinal values of risk exposure. The most frequent scale is low, medium, and high values. Four tools expand the scale to five and seven possible values, ranging from very low to very high.

As mentioned earlier, some tools are based on an existing risk assessment model. For example, the most common qualitative risk measurement used in the tools listed in Table 9 is from the ISO 27000 family and NIST CSF.

**Table 9.** Tools with qualitative risk measurement models.

| Tool | Description of the Measurement Method | Risk Scale |
|---|---|---|
| ADAMANT | It uses the ISO/IEC 27001:2013 risk measurement method | Very low, Low, Rather low, Medium, Rather high, High, and Very high |
| RACOMAT | It uses the ISO/IEC 27001:2013 risk measurement method | Low, Medium, and High |
| TISRIM | It uses the ISO/IEC 27001:2013 risk measurement method | Low, Medium, and High |
| AWWA | It uses the NIST CSF risk measurement method | Low, Medium, and High |
| CSET | It uses the NIST CSF risk measurement method | Low, Medium, and High |
| CSAT | It uses the NIST CSF risk measurement method | Low, Medium, and High |
| Patch management | It uses the NIST CSF risk measurement method | Low, Medium, and High |
| CSRAS | It uses the NIST 800-30 risk measurement method | Low, Medium, and High |
| Tritilanunt, Suratose tool | It establishes impact and probability risk values using the OSWAP risk level matrix | Low, Medium, High, and Critical |
| Asset assessment methodological tool | It uses the EBIOS risk measurement method considering (C) confidentiality, (I) integrity, and (A) availability | Low, Medium low, Medium, Medium high, and High |
| SGIS toolbox | It uses (C) confidentiality, (I) integrity, and (A) availability | Low, Medium, High, Critical, and Highly critical |
| OSCAD-Ciras tool | It uses the variables: probability of the event, consequences of the event, class of countermeasure, and countermeasure implemented | Low, Medium, and High |
| MVS visualization platform | It uses set theory to describe specific scenarios for each asset, vulnerability, and risk scenario using architectural risk analysis (ARA) | Low, Medium, and High |
| Eye retina | There is no information on how it performs the measurement | Low, Medium, and High |
| GFILanguard | There is no information on how it performs the measurement | Low, Medium, and High |
| N circle | There is no information on how it performs the measurement | Low, Medium, and High |
| Nessus | There is no information on how it performs the measurement | Low, Medium, and High |
| Nikto | There is no information on how it performs the measurement | Low, Medium, and High |
| OpenVAS | There is no information on how it performs the measurement | Low, Medium, and High |
| Security system analyzer | There is no information on how it performs the measurement | Low, Medium, and High |
| Qualys guard | There is no information on how it performs the measurement | Low, Medium, and High |
| Risk watch | There is no information on how it performs the measurement | Low, Medium, and High |
| SAINT | There is no information on how it performs the measurement | Low, Medium, and High |

ISO/IEC 27005:2018 provides recommendations on how to perform qualitative risk measurement. The two main variables are the probability of an incident scenario and the possible process impact if a threat were to exploit a vulnerability. Figure 4 shows each of the two main variables, weighted on a scale of five values ranging from very low to very high, where a numerical value indicates the relationship between the two variables. For example, if the numerical value of the above relationship is between 0 and 2, it is considered low risk; if it is between 3 and 5, it is considered medium risk; and if it is between 6 and 8, it is considered high risk.

| | Likelihood Of incident scenario | Very low (Very unlikely) | Low (unlikely) | Medium (possible) | High (likely) | Very high (frequent) |
|---|---|---|---|---|---|---|
| **Business impact** | Very low | 0 | 1 | 2 | 3 | 4 |
| | Low | 1 | 2 | 3 | 4 | 5 |
| | Medium | 2 | 3 | 4 | 5 | 6 |
| | High | 3 | 4 | 5 | 6 | 7 |
| | Very high | 4 | 5 | 6 | 7 | 8 |

**Figure 4.** Risk assessment according to ISO/IEC 27005:2018.

The NIST cybersecurity framework does not assess risks individually but conducts a global assessment of the organization. The NIST cybersecurity framework defines risk by tiers. The tiers are defined by NIST as levels of framework implementation. These tiers provide the context for how an organization perceives cybersecurity risks and how they can manage such risks. The tiers characterize the practices of an organization with the following values: partial (Tier 1), risk-informed (Tier 2), repeatable (Tier 3), and adaptive (Tier 4).

Other tools, such as the Tritilanunt and Suratose tools, perform risk measurement with another known method. For example, the Open-Web Application Security Project (OWASP) risk assessment model, which scores risk using the OWASP risk level matrix [48], is shown in Table 10.

**Table 10.** OWASP risk level matrix.

| Impact and Likelihood Levels | | | |
|---|---|---|---|
| **Likelihood Impact** | **Low** | **Medium** | **High** |
| High | Medium | High | Critical |
| Medium | Low | Medium | High |
| Low | Low | Low | Medium |

A group of tools (asset assessment methodological tool or SGIS toolbox) uses the three main attributes of cybersecurity (availability, integrity, and confidentiality) to measure the risk.

On the other hand, tools such as OSCAD-Ciras and MVS create a unique risk measurement considering specific variable relationships, such as the probability of occurrence and threats or set theory for specific assets or previously defined vulnerabilities.

The last group of tools needs to provide more information on how risk is measured. These tools (Nessus, SAINT, OpenVAS, Nikto, Eye retina, GFILanguard, N circle, security system analyzer, and Qualys guard) are focused on vulnerability analysis. They perform a vulnerability scan at a technical level. However, vulnerability analysis tools are subscription tools, and there needs to be more information available on how they perform risk measurement. The only information on the vulnerability analysis (subscription) tools is that they all give low, medium, and high values due to the scoring activity.

Derived from the analysis of qualitative tools, we can conclude that the variables most used to measure cybersecurity risks are impact and probability. Additionally, other variables, such as the three attributes of cybersecurity (integrity, availability, and confidentiality), must be considered to assess this impact and probability and, thus, establish a qualitative relationship between risk and these three attributes.

### 4.3.2. Quantitative Tools

A total of 12 tools perform risk measurement quantitatively. Table 11 shows the quantitative tools. They output a numerical risk value based on the input values of a set of variables (shown in Table 11). In addition, Table 11 shows either the underlying model or the type of mathematical model used in the Underlying Measurement Method column.

Quantitative tools consider different variables to qualitative tools because they serve different purposes. Qualitative tools are easier to use and are applied by companies with a lower level of maturity. However, one of their shortcomings is that they may need to be more accurate and objective. On the other hand, quantitative tools are usually used by organizations with a medium or high level of maturity where a higher level of accuracy and precision is required [49].

The variables used to measure the impact and probability of risk in a quantitative tool consider more specific variables than the qualitative tools, e.g., the range of occurrence, the risk exposure, or the internal rate of return. As they consider more specific parameters with complex relationships, these variables can quantify the risk numerically [49].

Quantitative tools usually consider already standardized algorithms to relate the risk measurement variables. The measurement method most used by the quantitative tools identified in the literature is the common vulnerability scoring system (CVSS) and the Bayesian attack graph (BAG).

**Table 11.** Quantitative risk measurement models in tools.

| Tool | Underlying Measurement Method | Input Variables for Risk Measurement |
|---|---|---|
| CRISM tool | Common vulnerability scoring system (CVSS) | The impact of exploiting the vulnerability, the asset values, how the vulnerability can be exploited, the complexity required for the vulnerability to be exploited, and how many instances of authentication are required to exploit the vulnerability. |
| Rimsha, tool | Common vulnerability scoring system (CVSS) | The damage assessment, the threat assessment, and the probability of the event. |
| Quantitative network security risk situation model | Common vulnerability scoring system (CVSS) | A BAG (Bayesian attack graph) tuple that considers the values of the following variables: threat capacity, vulnerability-exploiting probability metric, expected time for vulnerability exploits, expected time for vulnerability removals, and threat prediction (goal, path, probability, time). |
| RL-BAGS | Q-Learning model and SARSA learning | The asset value (AV), rate of occurrence (RO), risk exposure (RE), probability of compromise (POC), influence of function (IOF), and cost to patch (CTP). |
| Data Protection Impact Assessment tool | Joint risk and trust model (JRTM) | The adjusted probability, the vulnerability index, the adjusted impact, and the asset index. |
| J. Hu model | Markov model | The basic operation dimension, the vulnerability dimension, and the threat dimension. |
| Mobile risk calculator | SANS Institute quantitative risk analysis step by step | The annualized rate of occurrence (ARO), the annualized loss expectancy (ALE), and internal rate of return (IRR). |
| R. Riesco model | Semantic reasoning algorithm and web rules language (SWRL) | The residual risk of the threat, the decreasing value of the impact (severity), the probability of the threat before and after applying a countermeasure, and the impact (severity) of the threat when it materializes. |
| Teng. algorithm | Analytical hierarchy process (AHP) | The value of the asset, the value of the threat, the value of the vulnerability, the impact of a loss of assets, the degree of exposure of the assets, the control measures, and the consequences of breach of confidentiality, integrity, and availability. |
| CYRVM | Two algorithms: one for calculating the arithmetic mean of vulnerabilities and another for the factoring matrix | The probability, the number of vulnerabilities, the number of systems, the value of the impact or likelihood of the vulnerability of a system in a test set, the calculated value of impact or likelihood of system vulnerability, the maximum deviation of the calculated value and a test set. |
| OSINT | Simple relationships between variables | The numerical value relative to the importance of the information in the organization. |
| MST risk analysis | A recursive type of algorithm relating network nodes | The impact of a disruption being realized into a network traffic, and the dependency of asset. |

Note that the CVSS algorithm was designed to quantitatively identify vulnerabilities considering integrity, availability, and confidentiality attributes. These attributes are also considered in the qualitative measurement tools. On the other hand, the BAG is a graphical model for measuring risk. One of the variables considered by BAG is vulnerability, used in conjunction with the CVSS algorithm. Another variable that BAG uses is the probability of the risk, which is measured by adding the probabilities of random variables, such as the attack vector used by cybercriminals.

Additionally, other tools use algorithms that consider other variables or parameters, such as the annualized rate of occurrence (ARO), the annualized loss expectancy (ALE), and the internal rate of return (IRR) used by the SANS algorithm.

Likewise, some tools use algorithms that are not specific for risk measurement but can be adapted for cybersecurity risk assessment purposes. Some examples are algorithms

such as the Markov model, the semantic reasoning algorithm and web rules language, and the analytical hierarchy process (AHP). These algorithms are used together with variables such as the probability of occurrence and impact to measure risk.

Finally, some tools, such as the CYRVM and OSINT tools, use quantitative risk measurement algorithms created by the relationships between the variables defined by the authors. The variables used in the case of these algorithms are diverse. However, some standard variables considered by the other tool groups are asset value, vulnerability, probability of occurrence, and risk for these two tools.

Quantitative tools generally measure risk based on the probability of occurrence and impact parameters. However, they also usually consider other sub-parameters, which they relate to through pre-established algorithms.

Summarizing the results of this subsection: (1) The main measures used for the qualitative calculation of risk are probability and impact. (2) The main qualitative risk measurement values are "High", "Medium", and "Low". (3) The variables mainly used for quantitative risk calculation involve multiple relationships, as in the case of the CVSS algorithm of the NCTS or the ARO of the SANS.

### 4.4. Post-Pandemic Tendencies

The literature review was conducted on literature published up to 2021. After this period, in 2022, it was identified that new studies published, such as the study of Northern et al. [15], mention the use of version 3.1 of the CVSS algorithm. This new version of the algorithm automated by NIST considers variables such as exposure to exploits.

In addition, another 2022 study by Willing [50], which compiles a series of opinions from Chief Information Security Officers, mentions that the main types of risks to be considered in software automation are legacy software, remote access policies, DDoS attacks, phishing, and malware. To consider these new types of risks, it is relevant to consider variables such as exposure to exploits or threat intelligence [51].

Finally, international organizations such as ISO or NIST have updated their risk assessment models ISO 27001:2022 and NIST CSF V2 in 2022. These new models are not yet published in their final versions to the public, but it is necessary to consider them for future research branches of software assessment. These new models respond to the constantly evolving cybersecurity environment resulting from the COVID-19 pandemic.

### 5. Risk Assessment Automation Proposal

In this section, a cybersecurity risk assessment model is proposed. The objective of creating a new risk assessment model proposal (research work objective III and IV) is to solve the current problem of the models used in the automation mentioned in Section 4.3. This proposal is intended to be easily automated.

Different approaches, depending on the organization's maturity, were created to generate this risk assessment proposal: (1) the qualitative calculation approach and (2) the quantitative calculation approach. Additionally, it was intended that both approaches are for the general-purpose domain and cover all activities of the risk assessment stage. Therefore, the risk assessment, as mentioned above, is divided into three sub-stages and six activities which can be seen in Section 4.2.

The objective of creating the two different approaches to the proposed risk assessment model was to facilitate the adaptability and scalability of the model by organizations, as some organizations have a lower or higher level of maturity in performing a risk assessment.

The proposed variables of this model were created from the variables identified in the models (see Section 4.3, and Table S1: Variables summarizing) and the corresponding sub-stages and activities of the risk assessment. The variables created are as follows:

The following variables are proposed for risk identification sub-stage:

- Relevance of the Asset in the Process (RAP)—Activity 1: defined by the asset owner (see Table 12 and Table S2: Exposure values of the asset);

- Monetary Value of the Asset in Dollars (MVA)—Activity 1: only applies in the quantitative variant and is defined by the asset owner;
- Value of the Information Contained in the Asset (VICA)—Activity 1: only applicable in the quantitative variant and is defined by the asset owner;
- Economic Value of the Asset (EVA)—Activity 1: is a relation of the previous Activity 1 variables in both models (see Table 12);
- Countermeasure Maturity (CM)—Activity 2: related to the number of times the countermeasure has been effective;
- Countermeasure Effectiveness (CE)—Activity 2: only Countermeasure Maturity is used in the qualitative variant. In the quantitative variant, the difference of the impact multiplied by the maturity of the countermeasure is used (see Table 12);
- Available Asset Information (AAI)—Activity 2: directly related to the number of adverse events related to the asset with public information;
- Threat Value (T)—Activity 2: the variables Available Asset Information (AAI) and Value of the Vulnerabilities (V) are used in both approaches, qualitative and quantitative. The value of EVA is added to assign an economic value to the equation (see Table 12);
- Asset Exposure (AE)—Activity 2: for the qualitative variant, the relationship between Countermeasure Maturity (CM) and Value of the Vulnerabilities (V) is used. The equation defined by the SANS institute [52] (https://www.sans.org/white-papers/849/ (accessed on 15 October 2022)) is used in the quantitative approach;
- Value of the Vulnerabilities (V)—Activity 3: it is defined using the CVSS algorithm version 3. Programmed by the NIST (https://nvd.nist.gov/vuln-metrics/cvss/v3-calculator (accessed on 15 October 2022)). The same output is used for qualitative and quantitative approaches since the CVSS algorithm gives results from 1 to 10 [53], see Table S3: Threat value.

The following variables are proposed for the risk analysis sub-stage:

- Number of Occurrences (ON)—Activity 4: number of adverse events recorded in a year;
- Years Registered (YR)—Activity 4: only applies in quantitative approach and is defined by years of existence or registration of the asset and its incidents;
- Probability (ARO)—Activity 4: in the qualitative approach, only the number of occurrences (ON) is considered. The probability in the quantitative approach is the number of occurrences by year (ON/YR);
- Impact (IM)—Activity 4: direct relationship between Threat Value (T) and Asset Exposure (AE) (see Table 12, and Table S4: Impact value);
- Risk Exposure Value (R)—Activity 5: calculated as a direct ratio of impact (IM) and probability (ARO) (see Table 12 or Table S5: Risk exposure value).

The following variables are proposed for the risk evaluation sub-stage:

- Acceptable Risk Value (ARV)—Activity 6: the value defined by the organization is regularly lower than the current risk value;
- Residual Risk (RR)—Activity 6: the qualitative approach relates the risk exposure and the countermeasure maturity, while the quantitative approach relates the difference in the value of the risk exposure at two different time instants (see Table 12).

For more detailed information on the two approaches of the proposed risk assessment model, please refer to the following document, which was created to explain in detail the qualitative and quantitative relations: https://short.upm.es/14145 (accessed on 6 September 2022).

**Table 12.** Variables created for the cybersecurity risk assessment proposal.

| Variables | Qualitative Proposal | Quantitative Proposal |
|---|---|---|
| Relevance of the Asset in the Process (RAP) | Defined by the owner of the asset | RAP Low = 1, RAP Medium = 2, RAP High = 3 |
| Monetary Value of the Asset in Dollars (MVA) | | Proposed by the owner of the asset |
| Value of the Information Contained in the Asset in Dollars (VICA) | | Proposed by the owner of the asset |
| Economic Value of the Asset (EVA) | EVA = RPA | $EVA = (MVA + VICA)*RPA$ |
| Value of Vulnerabilities (V) | NIST algorithm CVSS: "Low" = 0–3.9 "Medium" = 4–6.9 "High" = 7–10 | V = CVSS quantitative version |
| Countermeasure Maturity (CM) | Low: Change or not effective Low: 0–3 times effective Medium: 4–8 times effective High: 9–10 times effective | CM = number of times the control/countermeasure has been effective (max. 10) |
| Countermeasure Effectiveness (CE) | CE = CM | $CE = (IM_{t-1} - IM_t)*CM$ |
| Asset Exposure (AE) | CM and V related by table | Percentage measure defined by the SANS institute model |
| Information Available on the Asset (AAI) | " Low" = Incidents < 1 per year "Medium" = Incidents >1, <2 per year "High" = Incidents > 2 per year | Number of incidents published per year |
| Threat Value (T) | AAI and V related by table | $T = [(V + AAI)/2]*EVA$ |
| Number of Occurrences (ON) | Low: 1 to 4 incidents per year Medium: 5 to 9 incidents per year High: 10 incidents or more per year | Number of negative events related to the asset with public information |
| Registered Years (YR) | | Years of existence of the asset |
| Likelihood (ARO) | ARO = ON | $ARO = \frac{ON}{YR}$ |
| Impact (IM) | T and AE related by table | $IM = T*AE$ |
| Risk Exposure Value (R) | IM and ARO related by table | $R_1 = IM*ARO$ $R_2 = [(T/EVA)*ARO]/2$ |
| Acceptable Risk Value (ARV) | Value of the risk immediately lower than the current one | Defined by the organization |
| Residual Risk (RR) | R and CM related by table | $RR = (R_t - R_{t-1})$ |

## 6. Validation of the Model Proposal and Results

In order to validate the proposed risk assessment model (research work objective IV), the steps described by Cabrero and Llorente were used to carry out a survey using individual aggregation [54]. During individual aggregation, information is obtained from experts, ensuring that the experts do not maintain contact with each other.

We spread the survey among the specialized forums of Reddit and LinkedIn. The experts were asked to evaluate the model through the survey.

This section is divided into two subsections: (1) validation of the proposal, where the survey is explained, and (2) results, where we present the survey results.

### 6.1. Validation of the Model Proposal

The objective of the questionnaire was to know the opinion of experts on the proposed risk assessment model presented in Section 5. Forty-two questions were created and divided into four sections as follows:

- Demographic questions: Eight questions related to the experts' basic information, such as the name of the organization where they currently work, the size of their organization, the country where they work, the industry related to their company, the maturity level of their organization, their current position, the number of years of experience in cybersecurity, and whether they have cybersecurity certifications;
- Risk management practices: Seven questions related to experts' and their organizations' current habits and practices regarding cybersecurity;
- Comments about risk assessment variables: 24 questions divided into six subcategories: (1) identification and evaluation of assets, (2) identification and evaluation of vulnerabilities, (3) threat identification, (4) impact, (5) likelihood, and (6) risk reduction. These subcategories aimed to obtain the experts' opinions on the general variables created for the risk assessment proposal;
- Evaluation of the model proposal: Three questions where the model proposal was formally presented in its two variants, qualitative and quantitative, and where the questionnaire asked for evaluation and feedback.

The questionnaire was created in two variants: (1) in the Spanish language and (2) in the English language. The questions of both questionnaires were the same, and both questionnaires were created with the Google Questionnaires tool and can be consulted in detail via the following links:

- Spanish: https://short.upm.es/tv2ci (accessed on 10 September 2022);
- English: https://short.upm.es/i6x90 (accessed on 30 September 2022).

Using cybersecurity forums on Reddit and LinkedIn as a point of contact, we sought personnel who meet the following requirements: (1) have at least three years of experience in cybersecurity or related area, (2) work in a medium to a large company, and (3) hold a manager, Senior, IT Auditor, Manager, or Director position. The questionnaire was sent to 50 experts working in various medium-to-large companies who meet the three previously established requirements. The SME definition used can be obtained from Annex I of the Regulation (EU) No. 651/2014 "SME Definition". Of 50 experts, 28 experts who met the previously established requirements voluntarily answered the survey in its entirety and met the criteria mentioned by Cabrero and Llorente [54].

The survey results can be consulted at the following link: https://short.upm.es/lofc5 (accessed on 30 November 2022). According to Cabrero and Lorette [54], the number of experts needed for a survey to be considered relevant must be less than 50 and greater than 15. Furthermore, the experts must meet a coefficient of argumentation and have adequate knowledge to participate in the questionnaire.

### 6.2. Results

After obtaining the responses of 28 experts, the results were consolidated in a single document that can be consulted at the following link: https://short.upm.es/wjvlz (accessed on 7 November 2022). Finally, they were analyzed to identify the following relevant results.

Demographic results are summarized in Table 13, and we can highlight that most of the experts who answered the survey have Senior Consultant or Manager positions. Additionally, over 90% have more than three years of experience directly related to information security or cybersecurity.

The experts also identified asset identification as the most relevant risk assessment sub-step, but they also consider all risk assessment sub-steps relevant. This tendency tells us that the experts consider that efforts should be invested in all the sub-stages to perform a risk assessment properly.

Another relevant point identified is the preference for quantitative risk models. Probably this preference is because experts are working in large companies with a medium–high level of maturity. In addition, they are familiar with or ready to perform more precise measurements to help detect possible security gaps.

**Table 13.** Demographic results.

| Relevant Expert Information | Answers |
|---|---|
| Position | 32% Senior Security Consultant<br>18% Manager<br>14% Security Consultant<br>11% IT Auditor<br>11% Security Supervisor |
| Experience | 93% More than 3 years<br>7% Less than 3 years |
| Certifications | 59% Yes<br>41% No |
| Company industry | 32% Professional services/consulting<br>21% Financial services<br>11% Technology<br>7% Manufacturing<br>7% Communications<br>7% Education |
| Size of companies | 96% Big companies<br>4% Medium companies |
| Security Maturity in Companies | 38% Advanced<br>46% High<br>4% Medium<br>8% Low<br>4% Null |

The survey was intended to be answered by a target audience defined as employees of large and medium-sized companies. Therefore, the target audience answered the survey. The survey was answered by employees, with 96% working in large companies and 4% in medium-sized companies. In addition, most of them have a high level of maturity in cybersecurity.

Finally, the diversity of sectors in which the experts work is varied, with the majority being in the professional services/consulting sector.

Derived from the demographic information, we can affirm that the target audience was adequate according to the steps of Cabrero and Llorente [54] and that the experts' opinion was objective, giving support and validity to their comments on the proposed cybersecurity risk assessment model.

The experts' answers about risk management practices are shown in Table 14, and we can highlight that the reference models most used by the experts coincide with the models identified in the SMR conducted in Section 3, since models such as the ISO 27000 family, NIST CSF, PCI-DSS, and NIST 800-30 are the models most used by the experts.

One area of opportunity identified is that most experts still need a tool that automates risk assessment, so a tool that facilitates quantitative and qualitative calculation would help to decrease the time and effort used for cybersecurity risk assessment.

A summary of remarks about risk assessment variables can be found in Table 15. This survey section was the main one because it evaluated the variables used in the proposed model.

Some relevant points are that the variables involved in quantifying the economic value of the assets considered relevant by the experts. However, not all use metrics to measure it in their respective organizations.

However, regarding assessing the exposure to exploits contemplated by the CVSS algorithm, almost 68% of the experts did not consider it relevant to measure to assess the exposure.

**Table 14.** Risk management practices.

| Relevant Expert Information | Answers |
|---|---|
| Most used models | 29.6% ISO 27000 family<br>14.8% COBIT 2019<br>14.8% Independent model<br>11.1% NIST CSF<br>7.4% NIST 800-30<br>7.4% MITRE<br>3.7% PCI-DSS |
| Risk assessment sub-stage relevance. | **Assets identification:**<br>46% Essential, 21% Very relevant, 29% Relevant, and 4% Almost irrelevant<br>**Threat identification:**<br>39% Essential, 32% Very relevant, 29% Relevant, and 0% Almost irrelevant<br>**Identification of vulnerabilities:**<br>39% Essential, 29% Very relevant, 32% Relevant, and 0% Almost irrelevant<br>**Identification of impact and probability:**<br>36% Essential, 28% Very relevant, 32% Relevant, and 4% Almost irrelevant<br>**Identification of countermeasures and calculation of residual risk:**<br>25% Essential, 39% Very relevant, 29% Relevant, and 7% Almost irrelevant |
| Type of calculation preferences | 61% Quantitative<br>49% Qualitative |
| Use of risk assessment tools | 52% No<br>48% Yes |

**Table 15.** Opinions about risk assessment variables.

| Subsection | Relevant Expert Information | Answers |
|---|---|---|
| Identification and evaluation of assets | Is the economic value of assets relevant?<br>Does your organization identify the economic value of assets?<br>Is the value of the information in the assets relevant? | 100% Yes, 0% No<br>75% Yes, 25% No<br>96% Yes, 4% No |
| Identification and evaluation of vulnerabilities | Is it important to measure impact of a vulnerability?<br>Consideration of availability, integrity, and confidentiality metrics<br>CVSS acceptation V3<br>Is it relevant to identify exploits exposure?<br>Use of exploit measures | 100% Yes, 0% No<br>93% Yes, 7% No<br>93% Yes, 7% No<br>96% Yes, 4% No<br>32% Yes, 68% No |
| Threat identification | Is the number of cybersecurity incidents related with an asset?<br>Use of cybersecurity incident metrics | 100% Yes, 0% no<br>82% Yes, 18% No |
| Impact | Is it relevant to measure the percentage of loss of assets after an attack?<br>Use of metrics to measure percentage of loss of an asset | 96% Yes, 4% No<br>50% Yes, 50% No |
| Likelihood | Is it relevant to measure the number of occurrences of security events?<br>Use of metrics for security occurrences<br>ARO (Number of occurrences/years) acceptance | 100% Yes, 0% No<br>71% Yes, 29% No<br>85% Yes, 15% No |
| Risk reduction | Is it relevant to identify countermeasure relevance?<br>Use of metrics for countermeasure effectiveness<br>Is it relevant to measure the residual risk?<br>Residual risk relation acceptance | 100% Yes, 0% No<br>71% Yes, 29% No<br>100% Yes, 0% No<br>89% Yes, 11% No |
| | Residual risk frequency | 29% Very frequent<br>36% Frequent<br>25% Sometimes<br>7% Almost never<br>3% Never |

All experts considered it relevant to value the number of incidents published on any technology or asset, but only 82% considered it as one of their organization metrics.

Likewise, 96% of the experts considered it relevant to value the percentage of loss of an asset, as performed by the SANS [52], while 50% only quantified the percentage loss of assets under attack.

Another relevant point is that the SANS Institute ARO variable was accepted by 85% of the experts as a good way of calculating probability.

Finally, all experts considered it relevant to measure the effectiveness of countermeasures and subsequently calculate residual risk. Most of them use some metric to measure the effectiveness of a countermeasure in their organizations. In addition, the ratio of the proposal was assessed by about 90% of the participants as adequate.

The final evaluation of the experts with the presentation of all the variables of the proposed cybersecurity risk assessment model is shown in Table 16. More than 90% of the experts agreed with the proposed quantitative and qualitative relationships. Likewise, the comments of the experts were generally positive. Therefore, it is considered that there was adequate acceptance of the proposal by the experts. The acceptance covers both the variables proposed in the quantitative and qualitative models.

**Table 16.** Evaluation of the proposal.

| Relevant Expert Information | Answers |
|---|---|
| Overall risk assessment proposal acceptance | 92.9% Yes<br>7.1% No |
| Experts; comments about proposal | Generally positive feedback and synchronizing the model's objectives with institutional priorities |

## 7. Conclusions and Future Work

While 25 primary studies were identified in the development of the systemic mapping review, it took time and effort to review all the literature. In addition, many studies were initially accepted due to their title and content but were subsequently removed because they were unrelated to cybersecurity.

Automations usually adapt an internationally accepted model or standard, such as the ISO 27000 family or the NIST CSF. A trend that relates automation to internationally accepted models was identified. The ISO 27000 family and the NIST CSF are the most accepted international models.

This trend is related to the desire of companies to convey confidence to competitors and customers. In addition, given the size of the companies, they usually already have a management system in place, most of the time created based on an international standard.

Organizations start with a qualitative calculation and then, with a higher level of maturity, move on to a quantitative analysis of cybersecurity risks. Due to this transition, most tools currently adopt qualitative calculation, as not all organizations have a level of cybersecurity maturity to adopt a quantitative risk assessment tool.

Additionally, related to the coverage of the tools, it was identified that eleven tools cover all the sub-steps of risk assessment. Activity 5 (risk determination) was the most covered by the tools, Activity 3 (vulnerability assessment) was the second most relevant to the tools, and, finally, Activity 6 (comparison of acceptable risk value and calculated risk) was the least considered by risk assessment tools.

A trend was also identified relating to vulnerability analysis tools being used to address risk reduction. This trend suggests that organizations are more interested in identifying vulnerabilities in technical assets than in other types of assets, such as personnel. In 100% of the cases, the experts considered it necessary to apply technical metrics to risk calculation, so the proposed risk assessment model considers exposure to exploits and other technical aspects included in the CVSS algorithm.

One of the sub-steps less frequently mentioned by the authors is related to risk decision making and establishing an acceptable risk value. This sub-stage is the one related to risk

treatment. This result could indicate a need for more information in the literature in this field. Comparing the opinion of the relevance of the risk assessment sub-stages with the information provided by the experts, we determined that it is considered a relevant sub-stage and that it is necessary to include it in risk assessment automation. Due to the effort and expense of fully automating the cybersecurity risk assessment stage, many tools focus on certain sub-stages or activities. Only eleven tools fully automate the entire cybersecurity risk assessment stage.

Related to the variables identified in the quantitative models, it is considered necessary to conduct further research on the calculation variables that can be used in quantitative approach tools. Experts considered quantitative calculation more relevant because, although most experts are familiar with qualitative methodologies, it is necessary to implement a higher level of maturity that can only be achieved using quantitative relationships.

Based on the trends described above, a risk assessment model was proposed that considers two variants (qualitative and quantitative) to serve as an adaptable starting point to improve the maturity and accuracy of risk calculation by considering variables that traditional models do not contemplate.

It was identified that the inclusion of the calculation of the economic value of assets, CVSS V3.1 algorithm (mentioned in the primary studies), the technical variables, the number of incidents, and the implementation of the probability using the SANS institute ratio (also mentioned in the primary studies) had a high acceptance by the experts. Furthermore, they evaluated the set of variables proposed as a valuable contribution to the field. Finally, the proposed model is intended to help in the lack of residual risk calculation models without modifying the concepts established by the authors of the primary studies. The experts accepted the incorporation of relationships for the calculation of residual risk and the effectiveness of countermeasures to solve the current problem in which this sub-stage of risk assessment is not given the necessary relevance.

This work is helpful as it provides an updated state of the art on the considerations of risk assessment stage automations. Additionally, variables considered by models of international organizations such as ISO, NIST, OWASP, etc., are identified and characterized. Finally, the model proposed in its two versions helps to carry out a risk assessment regardless of the size of the organization, considering the opinion and improvements presented by experts in the field of cybersecurity. This risk assessment proposal is easily automatable by considering variables and defined mathematical relationships.

In future work, we intend to carry out complementary studies related to other stages of risk management, such as risk treatment or monitoring and review, to identify gaps and possible contributions that may be useful for incorporating emerging cybersecurity risk management models.

**Supplementary Materials:** The following supporting information can be downloaded at: https: //www.mdpi.com/article/10.3390/app13010395/s1, Table S1: Variables summarizing; Table S2: Exposure values of the asset; Table S3: Threat value; Table S4: Impact value; Table S5: Risk exposure value (Impact and Probability); Table S6: Risk exposure value (Control maturity and risk).

**Author Contributions:** Methodology, T.S.F.G.; Writing—original draft, I.D.S.-G.; Writing—review & editing, J.M.; Supervision, T.S.F.G. All authors have read and agreed to the published version of the manuscript.

**Funding:** This research received no external funding.

**Data Availability Statement:** The information related to this manuscript can be consulted in a public and read-only manner from the publication of this article in the following links: 1—SMR process and relevant studies: https://short.upm.es/ytaub; 2—Risk assessment proposal: https: //short.upm.es/14145; 3—Risk assessment survey (Spanish): https://short.upm.es/tv2ci; 4—Risk assessment survey (English): https://short.upm.es/i6x90.

**Conflicts of Interest:** The authors declare that they have no known competing financial interests or personal relationships that could have appeared to influence the work reported in this paper.

# Appendix A

**Table A1.** Relevant studies before quality assessment.

| ID | Title | Year | QA1 | QA2 | QA3 | QA4 | QA5 | Total |
|----|-------|------|-----|-----|-----|-----|-----|-------|
| [8] | Fuzzy Application With Expert System for Conducting Information Security Risk Analysis | 2014 | 0.0 | 0.0 | 0.5 | 1.0 | 0.5 | 2.0 |
| [30] | Business Driven ICT Risk Management in the Banking Domain with RACOMAT | 2017 | 1.0 | 0.0 | 1.0 | 0.5 | 1.0 | 3.5 |
| [55] | Risk management practices in information security: Exploring the status quo in the DACH region | 2020 | 0.5 | 0.0 | 0.5 | 0.5 | 0.5 | 2.0 |
| [32] | Mobile Information Security Risk Calculator | 2019 | 1.0 | 0.0 | 1.0 | 1.0 | 1.0 | 4.0 |
| [28] | A risk assessment model for selecting cloud service providers | 2016 | 1.0 | 0.0 | 0.5 | 1.0 | 1.0 | 3.5 |
| [39] | A Web Platform for Integrated Vulnerability Assessment and Cyber Risk Management | 2019 | 1.0 | 1.0 | 1.0 | 1.0 | 0.5 | 4.5 |
| [31] | Open-source intelligence for risk assessment | 2018 | 0.0 | 1.0 | 1.0 | 1.0 | 0.5 | 3.5 |
| [40] | Development of Threat Modelling and Risk Management Tool in Automated Process Control System for Gas Producing Enterprise | 2019 | 0.5 | 0.0 | 0.5 | 1.0 | 0.5 | 2.5 |
| [41] | Automatic network restructuring and risk mitigation through business process asset dependency analysis | 2020 | 0.0 | 0.0 | 1.0 | 1.0 | 0.5 | 2.5 |
| [33] | Reducing Informational Disadvantages to Improve Cyber Risk Management | 2018 | 0.0 | 1.0 | 1.0 | 1.0 | 0.5 | 3.5 |
| [38] | CSAT: A User-interactive Cyber Security Architecture Tool based on NIST-compliance Security Controls for Risk Management | 2019 | 1.0 | 0.0 | 1.0 | 1.0 | 1.0 | 4.0 |
| [26] | Smart grid cybersecurity risk assessment | 2015 | 1.0 | 0.0 | 1.0 | 1.0 | 1.0 | 4.0 |
| [21] | Asset Assessment in Web Applications | 2010 | 1.0 | 0.0 | 0.5 | 1.0 | 0.5 | 3.0 |
| [27] | Security Assessment of Information System in Hospital Environment | 2016 | 1.0 | 0.0 | 0.5 | 1.0 | 1.0 | 3.5 |
| [25] | Experimentation tool for critical infrastructures risk management | 2015 | 1.0 | 0.0 | 0.5 | 1.0 | 1.0 | 3.5 |
| [24] | Sector-Specific Tool for Information Security Risk Management in the Context of Telecommunications Regulation (Tool Demo) | 2014 | 1.0 | 1.0 | 0.5 | 0.0 | 1.0 | 3.5 |
| [22] | A visualization and modelling tool for security metrics and measurements management | 2011 | 1.0 | 0.0 | 0.5 | 1.0 | 0.5 | 3.0 |
| [29] | A Comparison of Cybersecurity Risk Analysis Tools | 2017 | 0.5 | 0.0 | 1.0 | 0.5 | 0.5 | 2.5 |
| [43] | I-HMM-Based Multidimensional Network Security Risk Assessment | 2020 | 0.0 | 0.0 | 1.0 | 1.0 | 0.5 | 2.5 |
| [34] | Audit Plan for Patch Management of Enterprise Applications | 2018 | 1.0 | 0.0 | 1.0 | 0.5 | 0.5 | 3.0 |
| [44] | Calculated risk? A cybersecurity evaluation tool for SMEs | 2020 | 1.0 | 0.0 | 1.0 | 1.0 | 0.5 | 3.5 |
| [35] | Introduction of a Tool-based Continuous Information Security Management System: An Exploratory Case Study | 2018 | 1.0 | 0.0 | 0.5 | 1.0 | 0.5 | 3.0 |
| [45] | Tackle Cybersecurity and AWIA Compliance with AWWA's New Cybersecurity Risk Management Tool | 2020 | 1.0 | 1.0 | 1.0 | 0.5 | 0.5 | 4.0 |
| [23] | Introduction of a Cyber Security Risk Analysis and Assessment System for Digital I&C Systems in Nuclear Power Plants | 2013 | 1.0 | 0.0 | 1.0 | 0.0 | 0.5 | 2.5 |
| [42] | Leveraging cyber threat intelligence for a dynamic risk framework: {Automation} by using a semantic reasoner and a new combination of standards ({STIX}™, {SWRL} and {OWL}) | 2019 | 0.0 | 1.0 | 0.5 | 1.0 | 1.0 | 3.5 |
| [46] | Algorithm for quickly improving quantitative analysis of risk assessment of large-scale enterprise information systems | 2020 | 1.0 | 0.0 | 0.5 | 1.0 | 0.5 | 3.0 |
| [36] | RL-BAGS: A Tool for Smart Grid Risk Assessment | 2018 | 0.0 | 1.0 | 1.0 | 1.0 | 0.5 | 3.5 |
| [37] | Security risk situation quantification method based on threat prediction for multimedia communication network | 2018 | 0.0 | 1.0 | 0.5 | 1.0 | 0.5 | 3.0 |
| [56] | Threat Risk Evaluator: A Tool for Assessing Threat-Specific Security Risks in the Cloud | 2019 | 0.0 | 0.0 | 0.5 | 0.0 | 0.5 | 1.0 |
| [57] | Database Design for Threat Modelling and Risk Assessment Tool of Automated Control Systems | 2019 | 0.0 | 1.0 | 0.0 | 0.0 | 0.0 | 1.0 |
| [58] | Research and Implementation of Intelligent Substation Information Security Risk Assessment Tool | 2019 | 0.0 | 0.0 | 1.0 | 0.5 | 0.5 | 2.0 |
| [59] | Risk assessment of mobile applications based on machine learned malware dataset | 2018 | 0.0 | 0.0 | 0.5 | 1.0 | 0.5 | 2.0 |
| [60] | Simulation platform for cyber-security and vulnerability analysis of critical infrastructures | 2017 | 0.0 | 0.0 | 1.0 | 0.0 | 0.5 | 1.5 |
| [61] | An information security risk assessment algorithm based on risk propagation in energy internet | 2017 | 0.0 | 0.0 | 0.5 | 1.0 | 0.5 | 2.0 |
| [62] | Network security risk level estimation tool for information security measure | 2016 | 0.0 | 0.0 | 0.5 | 1.0 | 0.5 | 2.0 |
| [63] | Automatic security management of computer systems | 2015 | 1.0 | 0.0 | 0.5 | 0.0 | 0.5 | 2.0 |
| [64] | Fuzzy tool for conducting information security risk analysis | 2014 | 0.0 | 0.0 | 0.5 | 1.0 | 0.5 | 2.0 |
| [65] | Data model extension for security event notification with dynamic risk assessment purpose | 2013 | 0.0 | 0.0 | 0.5 | 1.0 | 0.5 | 2.0 |
| [66] | A multi-objective genetic algorithm for minimising network security risk and cost | 2012 | 0.0 | 0.0 | 0.5 | 1.0 | 0.5 | 2.0 |
| [67] | Information security risk reduction based on genetic algorithm | 2012 | 0.5 | 1.0 | 0.5 | 0.0 | 0.0 | 2.0 |

## Appendix B

**Table A2.** Articles refused after third screening.

| ID | Title | Year | Authors |
|----|-------|------|---------|
| [68] | Visualization process assisted by the Eulerian video magnification algorithm for a heart rate monitoring system: mobile applications | 2020 | A. Alarifi et al. |
| [69] | Technical, legal and ethical dilemmas: Distinguishing risks arising from malware and cyber-attack tools in the 'cloud'-a forensic computing perspective, | 2013 | V. Broucek and P. Turner |
| [70] | The impact of predicting attacker tools in security risk assessments, | 2010 | E. Gutesman and A. Waissbein |
| [71] | Static analysis for web service security—Tools & techniques for a secure development life cycle, | 2015 | A. Masood and J. Java |
| [72] | "The big data system, components, tools, and technologies: a survey | 2019 | T. R. Rao et al. |
| [73] | Online risk-based authentication using behavioral biometrics | 2014 | I. Traore et al. |
| [74] | Multi-criteria model for evaluation of information security risk assessment methods and tools | 2010 | M. Sajko et al. |
| [75] | Advances in Security and Privacy of Multimedia Big Data in Mobile and Cloud Computing | 2018 | B. B. Gupta et al. |
| [76] | Taking Compliance to the Cloud-Using ISO Standards (Tools and Techniques) | 2018 | T. Weil |
| [77] | SMSAD: a framework for spam message and spam account detection | 2019 | K. S. Adewole et al. |
| [78] | Risk-based thinking of ISO 9001:2015—The new methods, approaches and tools of risk management | 2017 | A. Y. Ezrahovich |

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
