# Peer review of "Cybersecurity Risk Assessment: A Systematic Mapping Review, Proposal, and Validation"

_applsci, doi:10.3390/app13010395_

Round 1

Reviewer 1 Report

PagePage 1 lines 32 to 36: a number of risk management model reference sources are referred to but more clarification is needed. They are not the same possibly and may need to be referred to later in the paper. So a clear argument is needed. It seems the information provided in the next paragraph links with the previous paragraph but it seems the two paragraphs need to be linked better/integrated more.

Page 3, last paragraph: a constructive criticism or appraisal is needed to evaluate the risk assessment substages and activities outlined.  How comprehensive are they? How useful are they? More depth is required.

In Section 3.1 it would be useful to cite the aim of the research and the research objectives. (The purpose of the research should appear in the abstract or the introduction section).

Section 3: where were the research questions drawn from? It is not clear!

Page 6: will the reader know what the Parsifal tool is? Why did the researchers select it?  

Is it possible to categorize the 51 publications accepted to allow the reader to establish what they were and where the knowledge was drawn from?  Are they academic journal articles (which ones) or conference papers (which ones)? The information in Table 4 possibly answers this question as the reference sources are cited. But more detail could be provided.

Section 4: Discussion and analysis. But should the analysis and then discussion. And where are the findings?

Table 6 needs to be explained in depth. Attention needs to be given to the material listed and an in-depth appreciation or assessment made. The information is only listed and what is the purpose? How does it link back to the claim in the abstract that the risk assessment models are mostly qualitative? All this needs to be looked into and explained. Arguments and value judgements need to be presented because it is not clear what the reader is supposed to learn from table 6.

Page 11: what type of risk? More detail is required. Can it be related to the type of industry for example and also, what is the link with cyber security provision? It is not clear how the risk assessment is identifying vulnerabilities to be fixed.

How were the experts selected?

How were the 28 key experts selected?

On reflection, is it possible to have better linkage and explanation of the subsections and how they are linked? In addition, it is not clear why the reader should read the paper. More interpretation is needed throughout and also, the conclusion needs to bring out the full contribution. The gap in the literature should be defined in the introduction and made clear in terms of the research outcome.

Possibly a contribution would be to show how the quantitative approach versus the qualitative approach to risk assessment can be used or to make a stronger case for one approach to proceed the other. But again, the research needs to be defined better and the research outcome clarified earlier in the paper. Hence, more tension needs to be given to the contribution made.

  1 lines 32 to 36: a number of risk management model 

Author Response

Comments from the reviewer #1

We are grateful to the reviewer for providing his/her suggestions. In the following section, all of these points have been addressed.

Comment #1:

Page 1 lines 32 to 36: a number of risk management model reference sources are referred to but more clarification is needed. They are not the same possibly and may need to be referred to later in the paper. So a clear argument is needed. It seems the information provided in the next paragraph links with the previous paragraph but it seems the two paragraphs need to be linked better/integrated more.

Response:

Thank you for your comment. In order to address this shortcoming, we have restructured the paragraph containing lines 32 to 36.

Comment #2:

Page 3, last paragraph: a constructive criticism or appraisal is needed to evaluate the risk assessment substages and activities outlined.  How comprehensive are they? How useful are they? More depth is required.

Response: We are grateful for your valuable suggestions. In connection with this comment, two paragraphs were added at the end of section 2 to reflect on the relevance of risk assessment sub-steps and activities.

Comment #3:

In Section 3.1 it would be useful to cite the aim of the research and the research objectives. (The purpose of the research should appear in the abstract or the introduction section).

Response: We are grateful for your valuable suggestions. To remedy this shortcoming, we have clarified and listed the 4 main objectives of the research work. In addition, these four objectives of the research work are taken up again in section three.

Comment #4:

Section 3: where were the research questions drawn from? It is not clear!.

Response: In order to address this shortcoming, in section 3 we have clarified that the research questions (RQ) refer to research objectives 1,2 and 3.

Comment #5:

Page 6: will the reader know what the Parsifal tool is? Why did the researchers select it?

Response: We are grateful for your valuable suggestion. In order to address this shortcoming, in section 3, we added why we use of the "Parsifal" tool. The tool was only used to reduce time and effort when analysing large amounts of data. Additionally, we included studies (Marchezan et al., 2019), (Karakan et al., 2020) and (Bustos Navarrete et al., 2018) that validate the use of the Parsifal tool in systematic mapping reviews. In these studies, the Parsifal tool is considered to be one of the most suitable tools to be used to support a systematic review and mapping without influencing the final results of an SMR.

Comment #6:

Is it possible to categorize the 51 publications accepted to allow the reader to establish what they were and where the knowledge was drawn from?  Are they academic journal articles (which ones) or conference papers (which ones)? The information in Table 4 possibly answers this question as the reference sources are cited. But more detail could be provided.

Response: We thank you for your valuable comments. We have added a link to the SMR process https://short.upm.es/ytaub, where supporting literature information is provided and have added more details on each step of the SMR process and the dataset used.

Comment #7:

Section 4: Discussion and analysis. But should the analysis and then discussion. And where are the findings?

Response:  We are grateful for your valuable observation. In order to address this shortcoming, we have added at the end of each part of section 4 the key sentence with findings.

Comment #8:

Table 6 needs to be explained in depth. Attention needs to be given to the material listed and an in-depth appreciation or assessment made. The information is only listed and what is the purpose? How does it link back to the claim in the abstract that the risk assessment models are mostly qualitative? All this needs to be looked into and explained. Arguments and value judgements need to be presented because it is not clear what the reader is supposed to learn from table 6.

Response: We thank you for your valuable comments. We have added the column on the type of tools (Qualitative or Quantitative), as well as two paragraphs that help to give more meaning and input to table 6.

Comment #9:

Page 11: what type of risk? More detail is required. Can it be related to the type of industry for example and also, what is the link with cyber security provision? It is not clear how the risk assessment is identifying vulnerabilities to be fixed.

Response: In order to address this shortcoming, we have completed the last paragraph of subsection 4.3 and added an additional paragraph with more details on the risks mentioned by the authors in the primary studies.

Comment #10:

How were the experts selected? And How were the 28 key experts selected?

Response: Thank you for your valuable comment. To remedy this shortcoming, we have added information to this section mentioning that the experts were selected on the basis of 3 criteria:

  • have at least three years of experience in cybersecurity or related area
  • work in a medium to a large company
  • hold a manager, Senior, IT Auditor, Manager, or Director position

In addition, information was added that they were contacted through specialized forums on Reddit and LinkedIn, where only 28 experts who met the 3 aforementioned criteria responded to the survey.

Comment #12:

On reflection, is it possible to have better linkage and explanation of the subsections and how they are linked? In addition, it is not clear why the reader should read the paper. More interpretation is needed throughout and also, the conclusion needs to bring out the full contribution. The gap in the literature should be defined in the introduction and made clear in terms of the research outcome.

Response: We appreciate your valuable comment. In response to this comment, we have clarified the research objectives in section I "Introduction" and completed the conclusions section to give greater emphasis to the research contribution.

Comment #13:

Possibly a contribution would be to show how the quantitative approach versus the qualitative approach to risk assessment can be used or to make a stronger case for one approach to proceed the other. But again, the research needs to be defined better and the research outcome clarified earlier in the paper. Hence, more tension needs to be given to the contribution made.

Response: To address this deficiency, we have added the relevance to the valuation of both approaches in the introduction and in several paragraphs in sections 4 and 6 in the hope of making it clearer that the contribution is to value both the qualitative and quantitative approaches.

Reviewer 2 Report

Suggest to add a section on the trends from 2018 to current. The cybersecurity scenario has changed tremendously since the pandemic. For the article to be more relevant what should cybersecurity professionals consider in terms of more sophisticated attacks like Ransomware. Authors can also add one section of what is the most prevalent attack since the pandemic and what can be done (include more tools).

Author Response

We are grateful to the reviewer for providing his/her suggestions. In the following section, all these points have been addressed.

Comment #1:

Suggest to add a section on the trends from 2018 to current. The cybersecurity scenario has changed tremendously since the pandemic. For the article to be more relevant what should cybersecurity professionals consider in terms of more sophisticated attacks like Ransomware. Authors can also add one section of what is the most prevalent attack since the pandemic and what can be done (include more tools).

Response: We appreciate your valuable comments. To remedy this deficiency, we have added subsection 4.4 Post-pandemic trends, which mentions relevant aspects of the variables identified in the previous subsection that have been updated in recent years.

Round 2

Reviewer 1 Report

The amendments are fine.